# p62 and NBR1 functions are dispensable for aggrephagy in mouse ESCs and ESC-derived neurons

Riccardo Trapannone[1,2] , Julia Romanov[1,2], Sascha Martens[1,2]

Accumulation of protein aggregates is a hallmark of various neurodegenerative diseases. Selective autophagy mediates the delivery of specific cytoplasmic cargo material into lysosomes for degradation. In aggrephagy, which is the selective autophagy of protein aggregates, the cargo receptors p62 and NBR1 were shown to play important roles in cargo selection. They bind ubiquitinated cargo material via their ubiquitin-associated domains and tether it to autophagic membranes via their LC3-interacting regions. We used mouse embryonic stem cells (ESCs) in combination with genome editing to obtain further insights into the roles of p62 and NBR1 in aggrephagy. Unexpectedly, our data reveal that both ESCs and ESC-derived neurons do not show strong defects in the clearance of protein aggregates upon knockout of p62 or NBR1 and upon mutation of the p62 ubiquitin-associated domain and the LC3-interacting region motif. Taken together, our results show a robust aggregate clearance in ESCs and ESC-derived neurons. Thus, redundancy between the cargo receptors, other factors, and pathways, such as the ubiquitin-proteasome system, may compensate for the loss of function of p62 and NBR1.

## Introduction

Mammalian cells require quality control systems to maintain their homeostasis, remove damaged components, and respond to various stresses. Macroautophagy (hereafter referred to as autophagy) entails the engulfment of cellular components by a double-membrane vesicle called autophagosome and the subsequent fusion of the autophagosome with lysosomes, wherein the cargo material is degraded (1, 2). In response to starvation, cells degrade cytoplasmic material in bulk to recycle cargo components for energy production and the synthesis of essential components. This catabolic process is referred to as bulk autophagy (3). In contrast, selective, cargo-driven autophagy can remove specific cargo material under nutrient-rich conditions (4). Selective autophagy

targets damaged organelles, protein aggregates, and infectious agents via cargo receptors, such as SQSTM1/p62, NBR1, NDP52, Optineurin, and TAX1BP1 (5). These proteins recognise cargoes that have been marked by the addition of ubiquitin chains for degradation. Afterwards, they recruit the autophagic starting machinery and trigger a series of events that lead to the nucleation and subsequent expansion of a membrane structure, called phagophore (6). The latter is modified via the covalent binding of ATG8 family proteins, such as LC3B. Finally, the autophagosome closes around the cargo and fuses with the endo-lysosomal compartment leading to the degradation of the cargo itself (7). This process is important for maintaining cellular quality control (8), and it is particularly relevant in cell types such as neurons, which cannot divide and therefore need a highly effective quality control to prevent cell death (9).

The accumulation of misfolded and aggregated proteins is a threat to cellular homeostasis. Various neurodegenerative pathologies, such as Alzheimer's disease, are associated with the accumulation of protein aggregates (10). The process by which ubiquitinated protein aggregates are removed by selective autophagy is called aggrephagy and it is driven by the cargo receptors p62, NBR1, and TAX1BP1 (11, 12, 13). p62 forms filamentous oligomeric structures via its N-terminal PB1 domain (14, 15). These oligomers are able to bind ubiquitinated substrates via the C-terminal ubiquitin-associated (UBA) domain (16). The recruitment of substrates by p62 leads to the formation of intracellular bodies, called condensates, which are removed by the autophagic machinery (11, 17, 18, 19). Mutations in the UBA domain cause the loss of ubiquitin binding and impair condensate formation in various cell lines (11, 20, 21). p62 also possesses an LC3-interacting region (LIR) motif, which is responsible for binding to phagophores modified with LC3 and GABARAP family proteins (22, 23). The region around the LIR motif additionally mediates the interaction with the scaffold protein FIP200, which recruits the autophagosome biogenesis machinery (24). Mutation of the LIR motif results in the accumulation and decreased autophagic degradation of p62 and decreased condensate formation (19, 23). p62 has been shown to play an important role in the clearance of toxic protein aggregates, such as Huntingtin and Tau (11, 25). Knockout of the *p62* gene in mice

[1]Max Perutz Labs, Vienna Biocenter Campus, Vienna, Austria   [2]Department of Biochemistry and Cell Biology, Center for Molecular Biology, University of Vienna, Vienna, Austria

Correspondence: riccardo.trapannone@univie.ac.at; sascha.martens@univie.ac.at

 

causes accumulation of ubiquitinated proteins and Tau aggregates in the brain ([26], [27]) and its overexpression reduces aggregate burden ([28]). The second aggrephagy cargo receptor, NBR1, does not oligomerise on its own, but it binds to p62 and cooperates with it in the recruitment of ubiquitinated substrates and the formation of condensates ([12], [19], [29]). Inhibition of lysosomal degradation by bafilomycin A leads to the stabilisation of p62 and NBR1 because they can no longer be degraded within lysosomes ([12]). TAX1BP1 is another cargo receptor that is recruited to p62-NBR1 condensates and is important for the degradation of protein aggregates ([13]). Although all of the three aggrephagy receptors are able to recruit the scaffold protein FIP200 and the autophagy initiation machinery, TAX1BP1 is the main factor linking the cargo and FIP200 for autophagosome biogenesis ([29]). Many aggrephagy cargo receptor studies have been performed in immortalised cell lines, making the interpretation of these results, with respect to their physiological roles, difficult. In addition, many cell biology data were generated by overexpressing one or more factors, potentially causing artifacts. Considering the crucial role that autophagy plays in the brain and its relevance in the protection from neurodegenerative diseases, neurons are excellent models for studying how cargo receptors cooperate and how mutations can affect the removal of ubiquitinated cargoes.

In this work, we used mouse embryonic stem cells (ESCs) to investigate the roles of the p62 and NBR1 cargo receptors in aggrephagy by introducing a fluorescent tag onto endogenous p62 via CRISPR/Cas9, and by investigating how p62 and NBR1 mutations and knockouts affect the formation of p62 condensates and aggregate clearance. ESCs can be differentiated into neurons, allowing us to compare cargo receptor dynamics between stem cells and their differentiated neuronal progeny. From this comparison, we showed that loss of function of the two cargo receptors p62 and NBR1 does not block the clearance of protein aggregates.

# Results

To image endogenous p62, CRISPR/Cas9 was used to add a streptavidin–mScarlet tag onto the endogenous *p62* gene in Rex1GFPd2 mouse ESCs. The newly generated cell line, referred to as mScarlet-p62 WT, could be differentiated into neurons, as confirmed by tubulin III-beta immunostaining (Fig S1A) and Western blot analysis (Fig S1B). Live cell imaging of DIV21 neurons showed condensate formation both in the cell body and in the processes (Fig S1C). These structures were also moving throughout the whole cell (Video 1, Video 2, and Video 3).

Next, we analysed the effect of an LIR motif mutation on p62 condensate formation and lysosomal delivery. It was previously reported that the mutation of the four residues in the LIR motif D335, D336, D337, and W338 of human p62 to four alanines disrupts the binding of p62 to LC3B and FIP200 ([22], [23], [24], [30]), therefore impairing the tethering of the cargo to the nascent phagophore and the recruitment of the autophagy initiation machinery, without affecting ubiquitin binding ([30]). We mutated the corresponding residues (D337, D338, D339, and W340) in the mScarlet–p62 WT cells

via CRISPR/Cas9 (Fig S1D–F, mSc-p62 LIR mut.). mScarlet–p62 LIR mutant cells were differentiated into DIV21 neurons, along with the parental mScarlet–p62 WT line (Fig S1E–G). LC3B lipidation in the mScarlet–p62 LIR mutant lines was comparable to the parental ones, suggesting that the genome editing procedure did not affect autophagy flux (Fig S1F). To visualise the degree of colocalisation of p62 with autophagosomes, we stained the cells with DAPGreen, a compound that emits fluorescence when it gets incorporated into autophagosomes and lysosomes ([31]). DAPGreen colocalised with overexpressed LC3B in HeLa cells (Fig S2A), with Rab7 (endosomes) in Hap1 cells (Fig S2B) and with LysoTracker (lysosomal dye) in Hap1 (Fig S2C) and mScarlet–p62 WT neurons (Fig S2D). Live cell imaging was performed so that p62- and DAPGreen-positive structures could be simultaneously observed (Fig 1A and B). Both p62 condensates and DAPGreen puncta were observed in cell bodies and processes in mScarlet–p62 WT and LIR mutant neurons (Fig 1A). 60% of the p62 puncta in the cell bodies and 80% of those in the processes were also positive for DAPGreen, showing that most of p62 is associated with autophagosomes and lysosomes in neurons (Figs 1B and S3A). Besides the punctate structures, the LIR mutant showed an increased diffuse cytoplasmic signal (highlighted in Fig S3B). This suggests that a larger part of the mutant p62 was not incorporated into condensates. In addition, the condensates in neuronal cell bodies were bigger when compared with the WT (Fig 1A and B, Video 4) and the protein levels of the LIR mutant were increased (Fig S1E), suggesting that the p62 LIR mutant is not turned over as efficiently as the WT form. However, the mutant p62 was still colocalising with DAPGreen (Fig S3A), demonstrating that the structures were still delivered into lysosomes.

To measure lysosomal turnover of WT and LIR mutant p62, we used ESCs instead of ESC-derived neurons because they tend to tolerate drug treatments better than neurons. This experimental setup also allowed us to compare how ESCs respond to mutations in the cargo receptors compared with neurons. Cells were treated with bafilomycin, to block lysosomal degradation, and/or with the proteasomal inhibitor MG-132 to determine the contribution of the proteasome to the degradation of p62. Cells were analysed by live cell imaging, showing that p62 puncta form in both WT and LIR mutant cells, although the mutant showed increased diffuse cytoplasmic localisation, as observed in the neurons (Figs 2A and S3B). Bafilomycin caused an increase in global p62 puncta number and size (Fig 2B), suggesting that both the WT and the LIR mutant p62 condensates are turned over within lysosomes. We also confirmed these data by Western blotting, probing for p62 and the cargo receptor NBR1. Although the difference between untreated and bafilomycin-treated samples is not as evident as in live cell imaging, both p62 and NBR1 band intensities tend to increase upon lysosomal inhibition (Fig 3A–C). LC3B lipidation takes place both in the WT and the LIR mutant cells, suggesting that the overall autophagic flux is not affected by the mutation of the cargo receptor (Fig 3A and B). Along with the LIR mutation, we also knocked out the whole *p62* gene in the background of the parental Rex1GFPd2 cells (Fig S1F). In this case, the NBR1 band was still slightly increased on average upon bafilomycin treatment and the general NBR1 levels were comparable to the WT line (Fig 3A and B). In addition, LC3B lipidation was still observed in the *p62* knockout

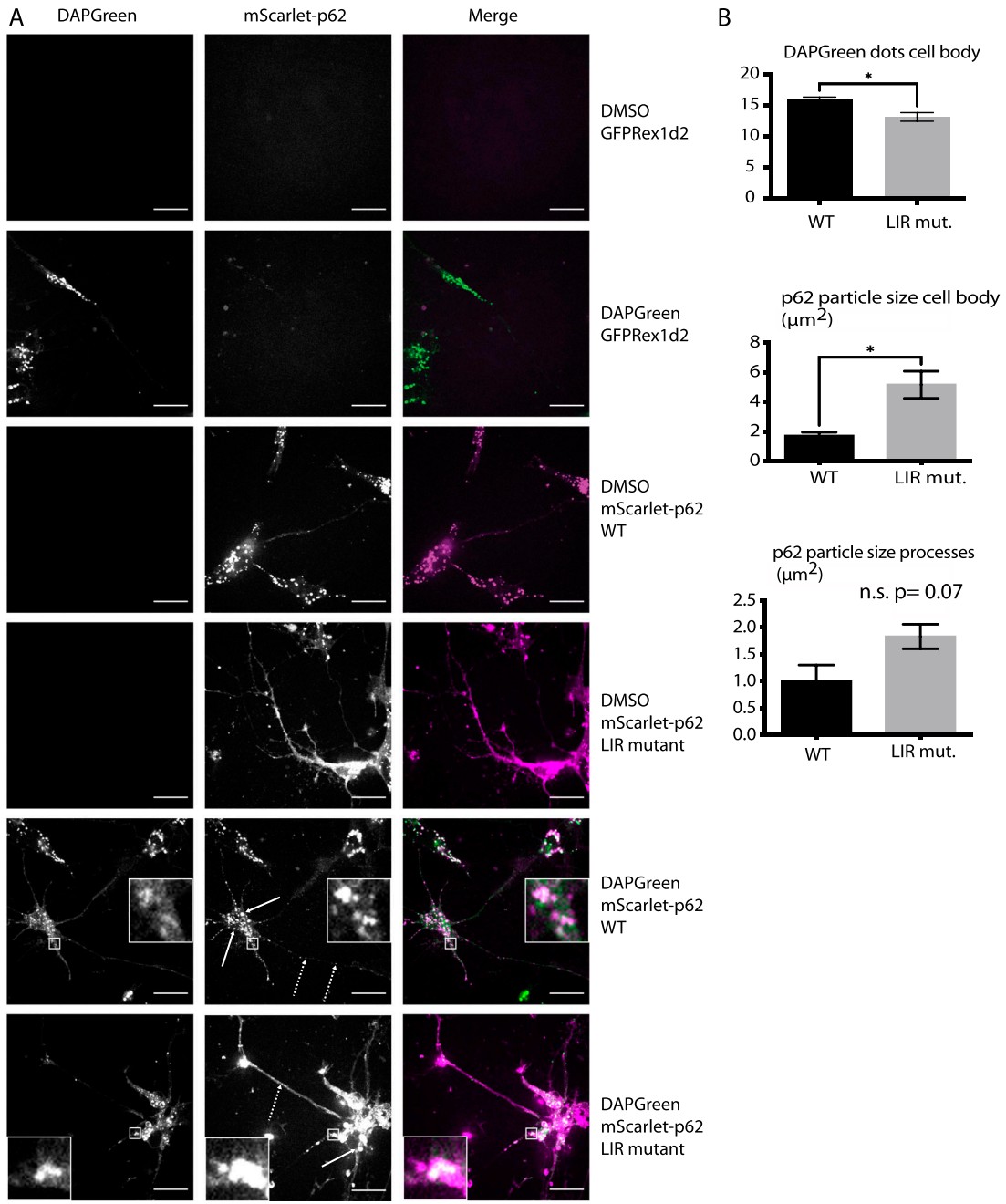

**Figure 1. Effect of p62 LC3-interacting region motif mutation on p62 condensate formation and endo-lysosomal colocalisation in neurons.**
**(A)** mScarlet–p62 WT or LC3-interacting region mutant DIV21 neurons were incubated with DAPGreen staining for 1 h and 30 min and observed via live cell imaging. Zoom factor of the selected regions: 5x. Scale bar: 20 μm. Standard or dashed arrows indicate p62 particles in the cell body or in the processes, respectively. **(B)** Quantification of the live cell imaging data. All the quantifications were performed on single stacks. The graph shows the mean ± SEM from n = 4 independent biological replicates. A total of 67–70 cell bodies and 8,818–8,839 μm of processes were analysed throughout the replicates. The P-value was calculated with a two-tailed t test, and the statistical significance is shown on the graph (P > 0.05 non-significant, 0.01 < P <0.05 *, 0.001 < P <0.01 **, P < 0.001 ***).

cell line (Fig 3A and B) and no difference was observed in *p62* knockout neurons (Fig S4A) and ESCs stained with DAPGreen compared with the WT (Fig S4B and C). Taken together, the data suggest that both in absence of p62 and when the p62 LIR motif is not functional, autophagy flux still proceeds under basal conditions.

The next question we asked was whether other domains of p62 might be important for condensate formation and turnover. It has been shown that a point mutation in the UBA domain (F406V) causes the loss of ubiquitin binding and the consequent lack of p62 condensate formation in HEK293 cells (21). Therefore, we investigated the effect of the UBA mutation by mutating the corresponding

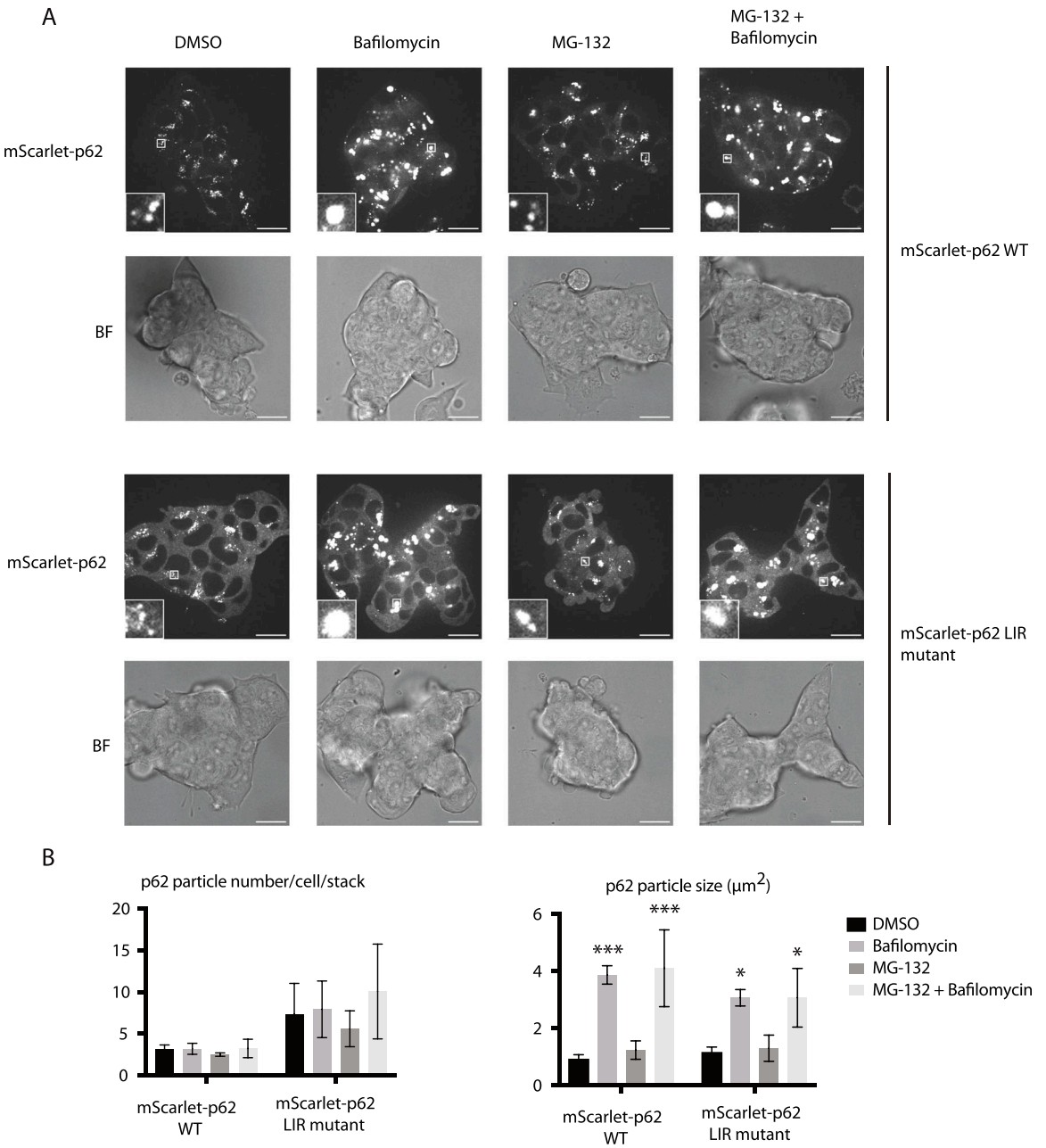

**Figure 2. Effect of the mutation in the LC3-interacting region motif of p62 in embryonic stem cells.**
mScarlet–p62 WT and mScarlet–p62 LC3-interacting region mutant embryonic stem cells were treated for 3 h with either 400 nM bafilomycin or 10 $\mu$M MG-132 or a combination of the two drugs. A DMSO-treated sample served as a negative control. **(A, B)** Live cell imaging analysis and (B) quantification of the microscopy data. The graphs represent the mean ± SEM from n = 3 independent experiments. Zoom factor of the selected regions: 5x. Scale bar: 20 $\mu$m. For each treatment and cell line, between 60 and 150 cells were analysed. The analysis was performed on a single-stack image. $P$-values were calculated with a two-way ANOVA and Tukey's multiple comparison test ($P > 0.05$ non-significant [not shown], $0.01 < P < 0.05$ *, $0.001 < P < 0.01$ **, $P < 0.001$ ***).

residue (F408V) via CRISPR/Cas9 in mScarlet–p62 WT cells (Figs S1F and S5A). Mutant cells were differentiated into DIV21 neurons alongside with the corresponding WT line (Fig S5B). Live cell imaging was performed to track p62 condensate formation and DAPGreen–p62 colocalisation. Surprisingly, no evident difference between the WT and the F408V mutant cells was observed under these basal conditions (Fig 4A and B). p62 condensates formed

normally, they still colocalised with DAPGreen and were transported normally in all directions (Video 5). We then tested whether this mutant form of p62 was indeed unable to bind ubiquitin using a microscopy-based pull-down assay. Incubation of lysates from mScarlet–p62 F408V DIV21 neurons with beads on which GST-4x ubiquitin chains were immobilised showed that the F408V mutation caused the loss of the interaction between p62 and ubiquitin chains

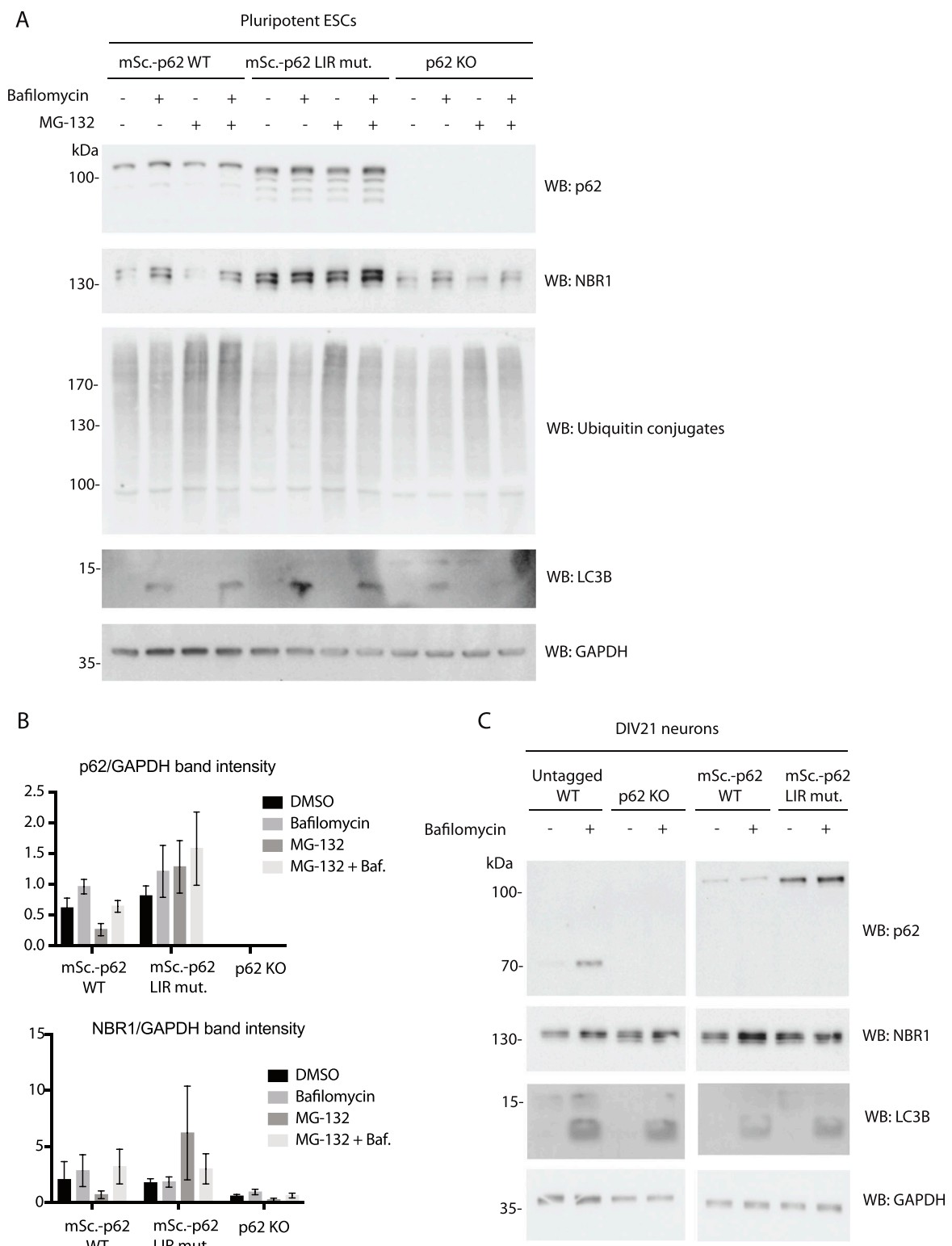

**Figure 3. Effect of p62 knockout and the mutation in the LC3-interacting region motif of p62 in embryonic stem cells and neurons.**
**(A)** mScarlet–p62 WT and mScarlet–p62 LC3-interacting region mutant embryonic stem cells were treated for 3 h with either 400 nM bafilomycin or 10 μM MG-132 or a combination of the two drugs. A DMSO-treated sample served as a negative control. Western blot analysis was performed using the antibodies indicated. **(A, B)** Quantification of the Western blot data from (A). p62 and NBR1 band intensity values were normalised to the GAPDH-loading control. The graphs represent the mean of the GAPDH-normalised values ± SEM from n = 3 independent experiments. P-values were calculated with a two-way ANOVA and Tukey's multiple comparison test (P > 0.05 non-significant [not shown], 0.01 < P < 0.05 *, 0.001 < P < 0.01 **, P < 0.001 ***). **(C)** DIV21 neurons were treated for 5 h with 400 nM bafilomycin. Cell lysates were analysed by Western blot.

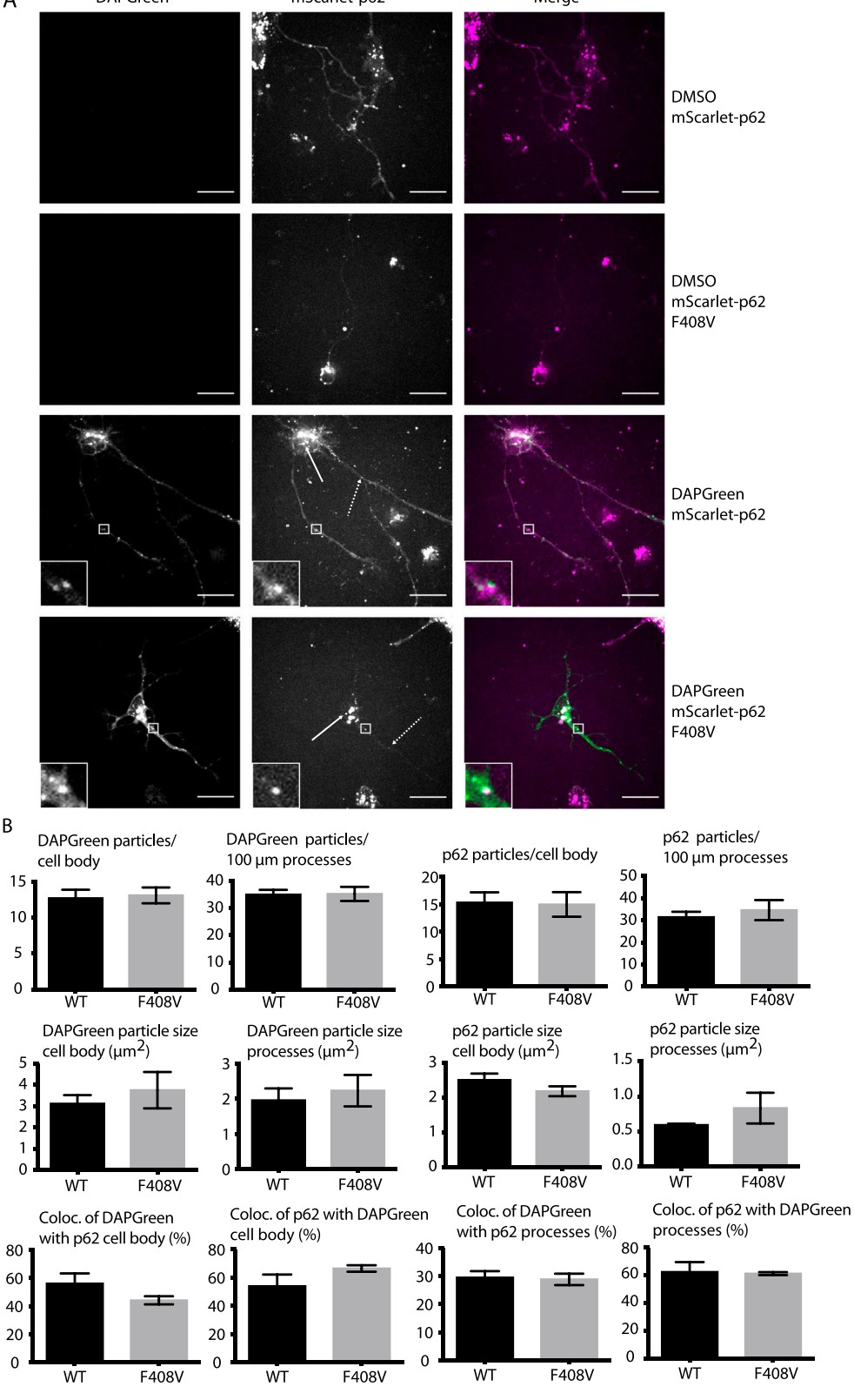

**Figure 4. Effect of p62 F408V mutation on p62 condensate formation and endo-lysosomal colocalisation in neurons.**
**(A)** mScarlet–p62 WT or mScarlet–p62 F408V mutant DIV21 neurons were incubated for 1 h and 30 min with DAPGreen staining and observed via live cell imaging. Zoom factor of the selected regions: 5x. Scale bar: 20 $\mu$m. Standard or dashed arrows indicate p62 particles in the cell body or in the processes, respectively. **(B)** Quantification of the live cell imaging data. All the quantifications were performed on single stacks. The graphs show the mean ± SEM from n = 3 independent biological replicates. A total of 55–60 cell bodies and 5,223–4,841 $\mu$m of processes were analysed throughout the replicates. The P-value was calculated with a two-tailed t test and the statistical significance is shown on the graph (P > 0.05 non-significant [not shown], 0.01 < P < 0.05 *, 0.001 < P < 0.01 **, P < 0.001 ***).

(Fig S6). Taken together, these data suggest that, although the F408V mutation abolished ubiquitin binding by p62, other cargo receptors may be able to take over this function. NBR1 was a likely candidate because it interacts with p62, co-localises with it in condensates, and provides a UBA domain to the complex (12, 29). We therefore knocked out NBR1 in either mScarlet–p62 WT or mScarlet–p62 F408V cells via CRISPR/Cas9 (Fig S7). The latter mutant lacks ubiquitin-binding by p62 and the NBR1 protein completely, but it was still able to differentiate into DIV21 neurons (Fig 5A). Unexpectedly, live cell imaging of these double-mutant neurons in presence of DAPGreen staining showed that p62 was still transported to DAPGreen-positive compartment, comparable to the cell line–expressing NBR1 (Fig 5A and B).

Next, we tested whether this finding would apply to the ESCs. p62 puncta formation was investigated in ESCs treated with MG-132 and bafilomycin. Under these experimental conditions, mScarlet–p62 WT cells were compared with the p62 F408V mutant line and to both of the NBR1 KO lines. Bafilomycin treatment caused increased p62 particle size in all of the mutants, suggesting that they are all able to deliver condensates into lysosomes (Fig 6A). Nevertheless, the size increase of the puncta observed after bafilomycin treatment was lower for all of the cell lines when compared with the WT ESCs (Fig 6B, right graph). This suggests that the mutations impaired the delivery of p62 condensates into lysosomes. The mutants were also analysed by Western blot probing for p62, NBR1, and LC3B (Fig 7A). A small increase in the p62 levels was observed upon bafilomycin treatment in the NBR1 KO cell line in presence of WT p62. Nevertheless, in the cell line harbouring the F408V mutation and deficient in NBR1, the delivery of the F408V mutant p62 into lysosomes appeared to be compromised, consistent with imaging data (Figs 6B and 7B). LC3B lipidation occurred normally, demonstrating that the autophagy flux was not generally affected. This was also observed in neurons treated with bafilomycin A1 (Fig 7C).

All of the experiments described so far were focused on p62 and NBR1 condensate formation under basal conditions. The next question was how these cells handle proteotoxic stress when ubiquitinated cargoes accumulate. A tool that can be used for tracking aggregate formation in cells is the puromycin pulse-chase assay. Puromycin inhibits ribosomal function and causes accumulation of truncated polypeptides (32). By exposing cells to puromycin, protein aggregates form and they can be monitored by an antibody specific for puromycin-modified peptides. The time needed for clearing out these puromycin aggregates is a good readout for the effect of mutations in the autophagy genes on the clearance of ubiquitinated polypeptides (13). All of the cell lines mentioned in this study were treated for 2 h with puromycin followed by a chase at different time points after its removal (Fig 8A–E). All of the cell lines were able to clear out more than half of the polypeptides after 6 h and most of the puromycin-positive signal was lost after 24 h. Every cell line was compared with the WT, and the quantifications of the Western blots (Fig 8F) showed that the recovery rate at every time point was comparable between cell lines and no major delay was observed. Interestingly, p62 levels increased after 24 h, suggesting that its expression was upregulated upon prolonged proteotoxic stress in WT, LIR mutant (Fig S8A), mScarlet–p62 WT/NBR1 KO (Fig S8B), mScarlet–p62 F408V (Fig S8C), and mScarlet–p62 F408V/NBR1 KO (Fig S8D) cells. The p62

KO cells served as a control (Fig S8E). The contribution of the proteasome and autophagy was also measured by adding the proteasomal inhibitor MG-132 or the autophagy inhibitor VPS34 IN1 (33) to WT cells in presence of puromycin. Samples treated for 6 h with a combination of treatments were analysed by Western blot, indicating that, although the proteasome may contribute to the puromycin-induced aggregate clearance, autophagy, blocked by VPS34 IN-1, still plays an important role in their removal (Fig S9A and B). The contribution of the proteasome and the autophagy machinery to aggregate clearance in these cells still requires more extensive investigation. Taken together, these results are in line with the data presented above showing that the mutations we introduced into p62 and NBR1 do not cause major changes to the autophagic flux and that the mutant cells were still able to remove protein aggregates after puromycin treatment.

## Discussion

Toxic protein aggregates represent a threat for cellular homeostasis and organismal health. Aggrephagy mediated by the p62 and NBR1 cargo receptors has been shown to contribute to the removal of these aggregates (reviewed in reference 34, 35, 36). In this study, we used mouse ESCs as a tool to investigate the role of the cargo receptors p62 and NBR1 in aggrephagy. These cells can be used in their pluripotent form or they can be terminally differentiated into cortical neurons. We tagged endogenous p62 with the fluorophore mScarlet using CRISPR/Cas9. We then tracked the endo-lysosomal compartments using the compound DAPGreen. p62 condensates were detected in all of the neuronal compartments (Fig S1) and these particles showed high colocalisation with the DAPGreen marker (Figs 1 and S3 and Video 1, Video 2, and Video 3). This was the first time p62 condensate dynamics were tracked in live neurons. Next, the LIR motif and the UBA domain of p62 were mutated. This way the protein lost the capacity to bind to the LC3/GABARAP family proteins on the phagophore and FIP200 and the ubiquitinated cargoes, respectively (21, 22, 23, 24). We also generated a cell line where the p62 gene was knocked out, and two lines where NBR1 was knocked out either on the background of the WT p62 or in presence of a ubiquitin-binding deficient form of p62. Interestingly, in both pluripotent cells and neurons, p62 was still delivered into the lysosomal compartment (Figs 1–7). This is apparently in contrast with some reports in immortalised cell lines, where the mutation of the LIR motif, and the UBA domain of p62, led to the loss of most of the p62 condensates (11, 19, 21). Nevertheless, we observed an increased p62 puncta size in neurons carrying the LIR mutation compared with the WT (Fig 1). Similarly, both p62 and NBR1 levels were higher in the LIR mutant in ESCs (Fig 3A) compared with both the WT and the p62 knockout cells. This observation suggests that the LIR mutation might affect the turnover of the two cargo receptors, for instance by keeping NBR1 trapped within the condensates. The mutation in the UBA domain and the NBR1 knockout did not cause any observable effect in neurons (Figs 4 and 5), but led to decreased lysosomal turnover of p62 in ESCs (Fig 6). In fact, the size of p62 condensates in these mutant cell lines was smaller compared with the WT cells in the presence of bafilomycin A1 (Fig 6B). This suggests that p62 F408V was not delivered into lysosomes

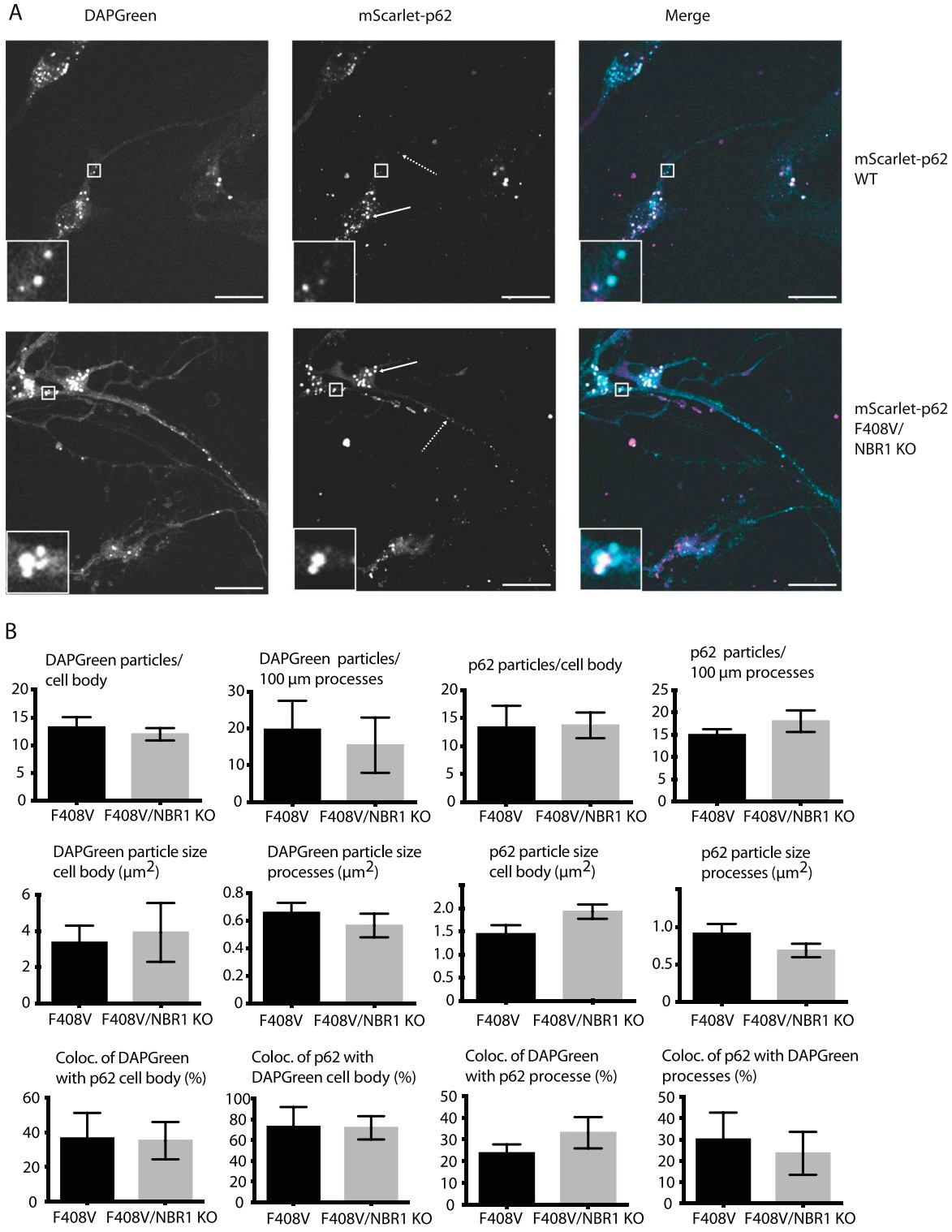

**Figure 5. Effect of NBR1 KO and simultaneous p62 F408V mutation on p62 condensate formation and endo-lysosomal colocalisation in neurons.**
**(A)** mScarlet–p62 WT or mScarlet–p62 F408V/NBR1 KO DIV21 neurons were incubated with DAPGreen staining for 1 h and 30 min and observed via live cell imaging (A). Zoom factor of the selected regions: 5x. Scale bar: 20 μm. Standard or dashed arrows indicate p62 particles in the cell body or in the processes, respectively. **(B)** Quantification of the live cell imaging data. All the quantifications were performed on single stacks. The graphs show the mean ± SEM from n = 3 independent biological replicates. A total of 49–50 cell bodies and 5,171–8,354 μm of processes were analysed throughout the replicates. The P-value was calculated with a two-tailed t test, and the statistical significance is shown on the graph (P > 0.05 non-significant [not shown], 0.01 < P < 0.05 *, 0.001 < P < 0.01 **, P < 0.001 ***).

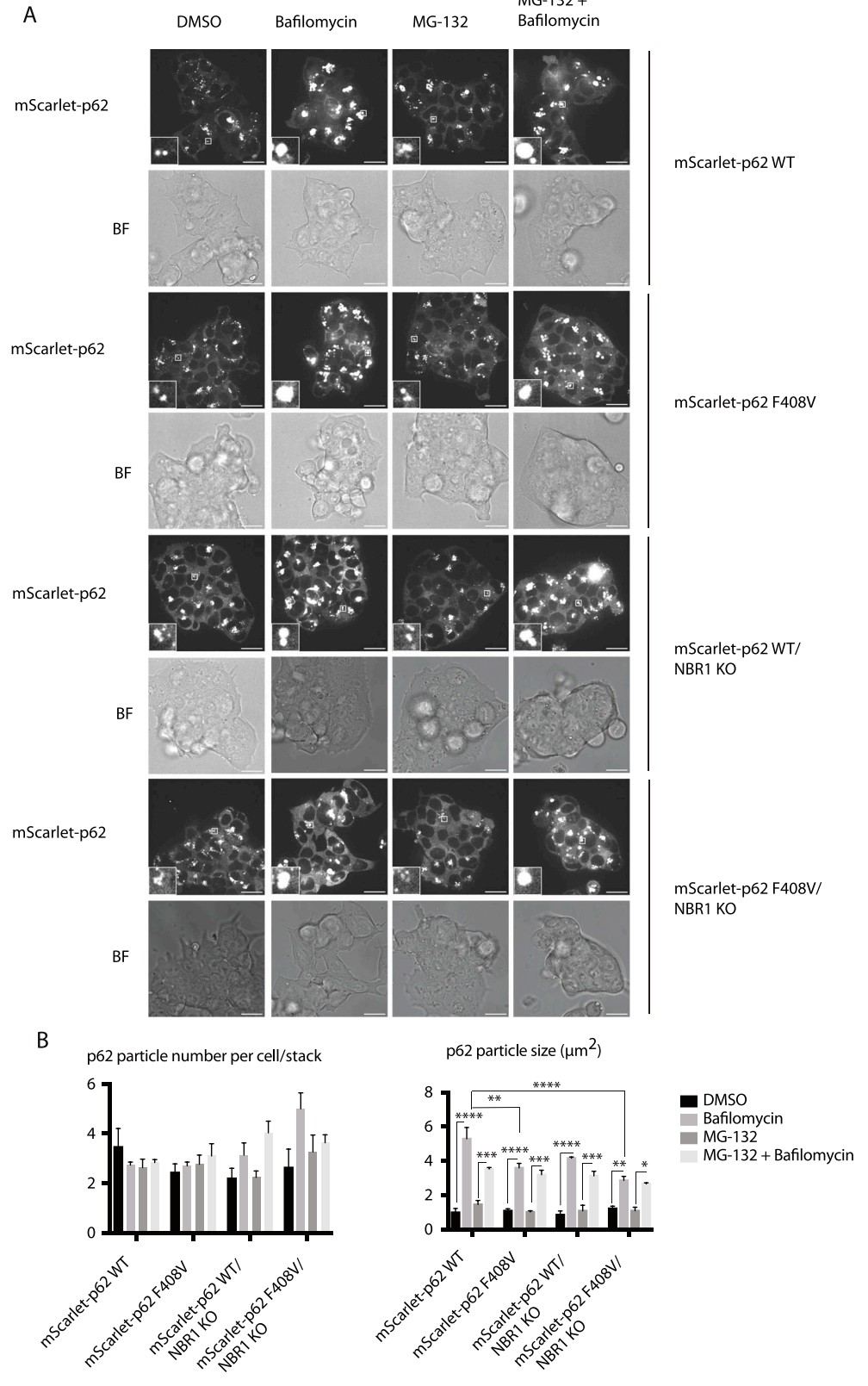

A

mScarlet-p62 / BF rows across DMSO, Bafilomycin, MG-132, MG-132 + Bafilomycin

mScarlet-p62 WT

mScarlet-p62 F408V

mScarlet-p62 WT/ NBR1 KO

mScarlet-p62 F408V/ NBR1 KO

B

p62 particle number per cell/stack

p62 particle size (µm²)

DMSO
Bafilomycin
MG-132
MG-132 + Bafilomycin

Figure 6.   Effect of NBR1 KO and p62 F408V mutation on p62 condensate formation in embryonic stem cells.

mScarlet–p62 WT, mScarlet–p62 F408V, mScarlet–p62 WT/NBR1 KO, and mScarlet–p62 F408V/NBR1 KO embryonic stem cells were treated for 3 h with either 400 nM bafilomycin or 10 µM MG-132 or a combination of the two drugs. A DMSO-treated sample served as a negative control. **(A, B)** Live cell imaging analysis and (B) quantification of the microscopy data. The graphs represent the mean ± SEM from n = 3 independent experiments. For each treatment and cell line, between 60 and 150 cells were analysed per replicate. The analysis was performed on a single-stack image. $P$-values were calculated with a two-way ANOVA and Tukey's multiple comparison test ($P > 0.05$ non-significant [not shown], $0.01 < P < 0.05$ *, $0.001 < P < 0.01$ **, $P < 0.001$ ***).

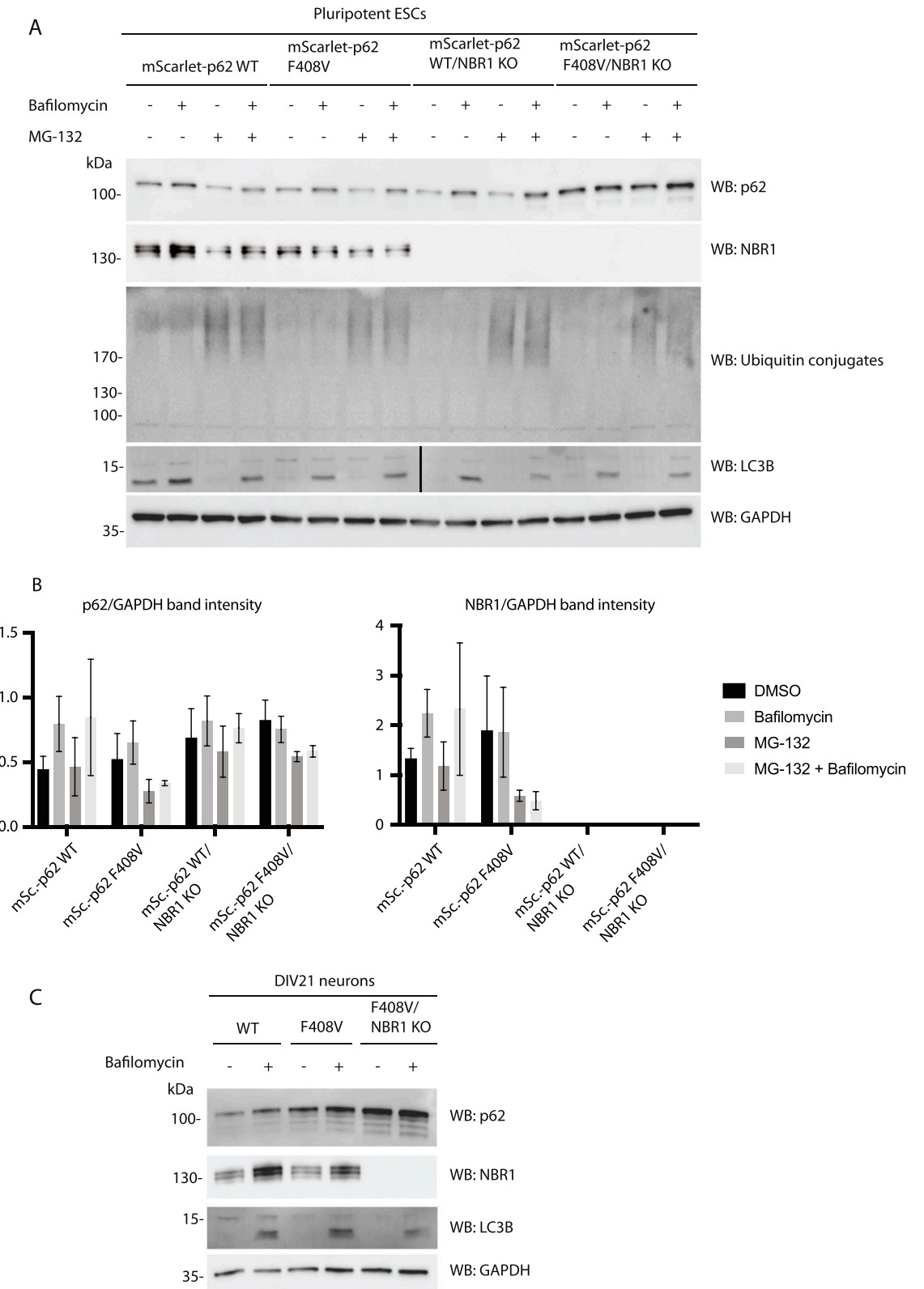

**Figure 7. Mutation of the p62 ubiquitin-associated domain (F408V) and knockout of NBR1 in embryonic stem cells and neurons.**
**(A)** mScarlet–p62 WT, mScarlet–p62 F408V, mScarlet–p62 WT/NBR1 KO, and mScarlet–p62 F408V/NBR1 KO embryonic stem cells were treated for 3 h with either 400 nM bafilomycin or 10 $\mu$M MG-132 or a combination of the two drugs. A DMSO-treated sample served as a negative control. Western blot analysis was performed using the antibodies indicated. **(A, B)** Quantification of the Western blot data from (A). p62 and NBR1 band intensity values were normalised to the GAPDH-loading control. The

as efficiently as WT p62, especially when NBR1 was knocked out on top. WT p62 was also not turned over efficiently in the lysosomes when NBR1 was absent.

It is essential for stem cells to maintain genomic and proteomic stability both when the cells are pluripotent and during differentiation. This ensures the correct transmission of the genetic information to the daughter cells and the differentiation into specialised cell types (37). Neurons, in particular, are more sensitive to proteotoxic stress and the failure to maintain their homeostasis leads to neurodegeneration. In fact, some autophagy genes are up-regulated in neurons (38) and their knockout leads to cell dysfunction and eventually death (39, 40, 41). This up-regulation is likely to make the system more stable and resistant to stress conditions and mutations. This explains why some of the mutations we introduced in this study had little or no effect in neurons but led to some phenotypes in ESCs. The observation that the LIR and the UBA mutants of p62 are still able to form condensates and are delivered to lysosomes, although with lower efficiency, suggests a redundancy between cargo receptors. Most notably, the other receptors that also possess LIR motifs might take over some p62 functions and bind the forming autophagosomes and the auto-phagic initiation machinery (34, 42). In the case of the UBA domain mutation, we hypothesise that, besides the cargo receptor re-dundancy, there are additional ways to recruit cargoes for auto-phagy in a ubiquitin-independent manner. This is suggested by the fact that, in the absence of NBR1 and with a defective p62 UBA domain, pluripotent cells and neurons are still able to process condensates and to clear out puromycin-induced misfolded pro-teins (Figs 5–8). Ubiquitin-independent cargo recruitment by p62 has already been shown in a few cases (28, 43). Moreover, the ZZ domain of p62 has been shown to bind to N-terminally arginylated proteins (44, 45), suggesting that there are different ways by which protein cargoes can be sequestered by p62. A ubiquitin-independent cargo recruitment mechanism has also been shown in mitophagy (46). Similar to what we observed with the point mu-tations, the removal of p62 and NBR1 in our system showed mild phenotypes with respect to the clearance of puromycin-induced aggregates. When NBR1 is knocked down in HeLa cells, p62 still form aggregates and vice versa, although the condensate formation is less efficient (12). Along the same lines, each cargo receptor be-comes stabilised by bafilomycin when the other one is absent, clearly showing that they are turned over within the lysosome independently from each other (12). Besides the redundancy of the two cargo receptors, both stem cells and neurons might possess additional factors that contribute to make their quality control even more robust and resistant to stress compared with cancer cell lines. In fact, none of the mutations described in this work had a strong effect on the clearance of puromycin-induced aggregates (Fig 8). It is clear that stem cells can prevent accumulation of toxic com-ponents even when the two main aggrephagy receptors p62 and NBR1 are not fully functional. Micro- or chaperone-mediated autophagy might also contribute to the degradation of toxic

aggregates (47, 48). In addition, a CCT2-dependent macroautophagy pathway may compensate for the loss of p62 and NBR1 function (49).

In conclusion, we have investigated the roles of p62 and NBR1 in aggrephagy in mouse pluripotent ES cells and in neurons via CRISPR/Cas9 manipulation of the endogenous genes. Working with endogenous protein levels and with cell lines that have properties similar to physiological conditions allowed us to better understand how the autophagic process is regulated. Our data show a robust autophagic regulation when the two main aggrephagy cargo receptors are not fully functional. Our work helps obtain a clear comprehension of how the cargo receptors recruit and remove potentially toxic cargoes. Such understanding is essential for unravelling the molecular basis of protein aggregate accumulation in diseases such as Alzheimer and Parkinson.

# Materials and Methods

### Mouse ESC culture

Rex1GFPd2 mouse pluripotent ESCs (50), were obtained from the laboratory of Dr Martin Leeb (Max Perutz Labs). The cells were grown at 37°C with constant 5% $CO_2$ supply and split at regular intervals with a typical splitting ratio ranging between 1:10 and 1:20. All the pluripotent lines were grown in flasks coated with 0.2% porcine gelatine in DMEM high-glucose (Sigma-Aldrich), supplemented with 1% penicillin/streptomycin, 1% GlutaMAX, 1 mM sodium pyruvate, 0.1 mM non-essential amino acids, 15% FBS, 55 mM 2-ME, 10 ng/ml leukaemia inhibitory factor (Protein Technologies Facility; Vienna BioCenter), 1.5 $\mu$M CHIR99021 (Axon Medchem), and 0.5 $\mu$M PD0325901 (Axon Medchem). All the cell lines were regularly screened for my-coplasma infection.

### Generation of the CRISPR cell lines

For generating the mScarlet–p62 WT cell line, Rex1GFPd2 cells were transfected with two plasmids: one encoding for the two guide RNAs, the Cas9 nickase and a puromycin resistance gene, the other carrying the homologous repair template. The transfection was performed using Lipofectamine 2000, according to the manufac-turer's instructions, with a ratio of 1:1 between the two plasmids. After 24 h, transfected cells were selected by the addition of fresh medium containing 5 $\mu$g/ml puromycin. The selection was kept for 72 h in total. Afterwards, the cells were placed back in medium without puromycin for a few days until colonies became visible. All the selected cells were detached and FACS-sorted into a pool based on the red fluorescence signal. All the gated cells were let recover for 3 d, detached and then FACS-sorted once more into 96-well plates so that each well contained a single cell. Clones became visible after 1 wk with an approximal survival rate of 30%. At this

---

graphs represent the mean of the GAPDH-normalised values ± SEM from n = 3 independent experiments. *P*-values were calculated with a two-way ANOVA and Tukey's multiple comparison test (*P* > 0.05 non-significant [not shown], 0.01 < *P* <0.05 *, 0.001 < *P* < 0.01 **, *P* < 0.001 ***). **(C)** DIV21 neurons were treated for 5 h with 400 nM bafilomycin. Cell lysates were analysed by Western blot.

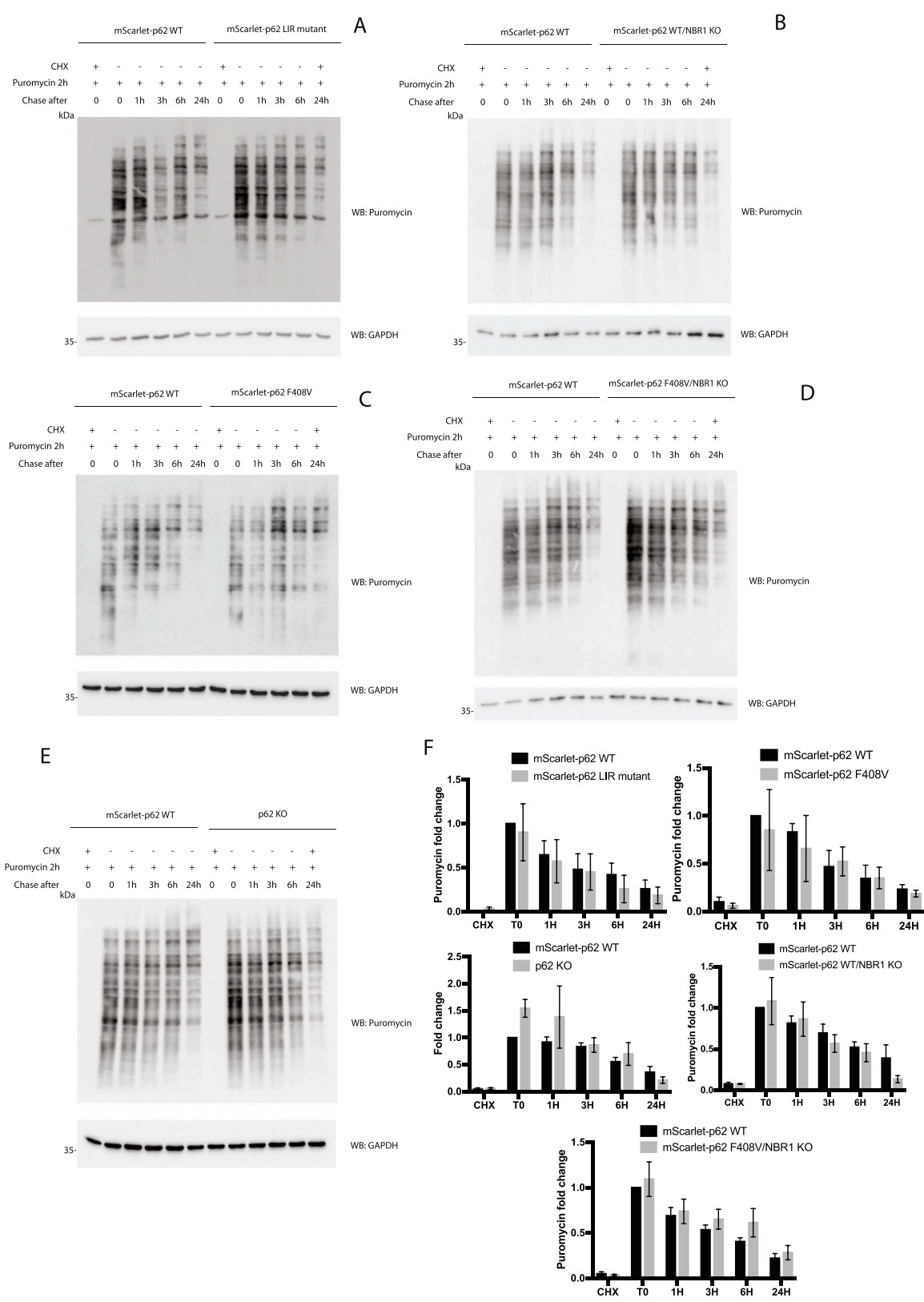

**Figure 8. Puromycin pulse–chase experiment showing the effect of p62 mutations, p62 knock-out, and NBR1 knock-out in embryonic stem cells.**
Cell lines were treated with 5 μg/ml puromycin for 2 h and either lysed immediately (T0) or let recover in absence of puromycin for the indicated time. A sample was pre-treated with 15 μM cycloheximide for 10 min and served as a negative control. Lysates were analysed by Western blot. **(A, B, C, D, E)** Comparison between mScarlet–p62 WT and mScarlet–p62 LC3-interacting region mutant (A), mScarlet–p62 WT/NBR1 KO (B), mScarlet–p62 F408V (C), mScarlet–p62 F408V/NBR1 KO (D), and p62 KO (E).

point, all the colonies were detached from each well, expanded, and analysed. For generating the mScarlet–p62 LIR mutant and the F408V mutant, mScarlet–p62 WT cells were used as parental cell line. The transfection and selection were performed as described above. A restriction site was introduced via a silent mutation in close proximity to the PAM site. Positive clones were selected based on the restriction enzyme cut on the PCR products.

For generating the KO cell lines, the following parental cell lines were used: Rex1GFPd2 for generating the p62 KO, mScarlet–p62 WT, and mScarlet–p62 F408V for the two NBR1 KO lines. Cells were transfected with one plasmid encoding the two guide RNAs, the Cas9 nickase, and a puromycin resistance gene. The transfection was performed as described above. After selection, cells were grown and then sorted into 96-well plates. Clones were then selected based on the PCR product size and expanded. For choosing the clone to be used for every cell line, cells were examined for the expression of the protein of interest and selected for expression levels similar to the WT in case of tag inserts or mutations and for no expression in case of knockout. Clones that showed unintended Cas9 integration were excluded.

### Neuronal differentiation

ESCs were differentiated using established protocols ([51], [52], [53]) with some minor modifications. On the splitting day, ESCs were detached, diluted in growing medium, and counted. 300,000 cells (for 10-cm dishes) and 45,000 cells (six-well plates) were seeded into tissue culture vessels, previously coated with 0.2% porcine gelatine for 30 min at RT. 24 h later, following two washes in PBS, DDM medium was added: DMEM-F12 (Sigma-Aldrich) supplemented with 1% penicillin/streptomycin, 1% GlutaMAX, 0.1 mM 2-ME, 0.1 mM non-essential amino acids, 1 mM sodium pyruvate, 0.05% BSA (Thermo Fisher Scientific), 1% N2 supplement (Thermo Fisher Scientific). This is defined as day in vitro 0 (DIV0). From DIV2 to DIV8, the medium was changed every second day and fresh DDM medium containing 1 μM cyclopamine (AdipoGen) was added each time. Dead cells were washed away with PBS before medium exchange when necessary. On DIV10, fresh DDM without cyclopamine was added after a gentle wash. On DIV12, the pre-differentiated cells were re-plated onto fresh plates, previously coated with a mixture of poly-L-lysine (33 μg/ml; Sigma-Aldrich) and laminin (3 μg/ml; Sigma-Aldrich) for 2 h at RT. 1 h before the detachment, DIV12 cells were gently washed with PBS and shifted into N2B27 medium, which is a 1:1 mixture of DMEM-F12 and neurobasal medium (Thermo Fisher Scientific), supplemented with 1% penicillin/streptomycin, 1% GlutaMAX, 0.05 mM 2-ME, 0.025% BSA, 0.5% N2 supplement, 1% B27 supplement (Thermo Fisher Scientific, without vitamin A), and 0.5 mM valproic acid (Sigma-Aldrich, optional). The coating mixture was washed away twice with PBS and let dry. At this point, the cells were washed, detached with Accutase (Sigma-Aldrich) for 4 min at 37°C, gently pipetted up and down 10–15 times with a 1-ml pipette, diluted in un-supplemented neurobasal medium and counted. The

suspension was spun once, resuspended in N2B27 medium by gentle pipetting (3–5 times with a 1-ml pipette), and diluted to the desired density. Cells were finally seeded: $5.6 \times 10^5$ cells/well were plated onto six-well plates and a higher density was used in 10-well live cell imaging slides ($0.4–0.5 \times 10^5$ cells/well). Fresh N2B27 medium without valproic acid was replaced every second day, starting from DIV14.

### Microscopy experiments

ESCs or neurons were grown on acid-treated glass coverslips, fixed in 4% PFA for 20 min at RT, permeabilised with 0.1% Triton X-100 for 5 min and incubated with blocking solution (1% BSA) for 1 h. Primary antibodies, diluted in blocking solution were added for 1 h. Afterwards, fluorophore-labelled secondary antibodies were added for another hour. Gentle PBS 1X washes were performed at the end of each step. The following antibodies and dilutions were used in this study: mouse anti-tubulin III-beta (1:1,000; BioLegend TUJ1), goat anti-mouse Alexa Fluor 647 conjugate (1:500; Thermo Fisher Scientific). Finally, the coverslips were mounted onto glass slides using DAPI-Fluoromount-G (SouthernBiotech). Images were acquired at the Zeiss LSM 980 confocal microscope, using a Plan-Apochromat 63x/1.4 Oil DIC M27 objective, 405 nm (30 mW), 488 nm (30 mW) and 561 nm (25 mW) laser diodes.

For live cell imaging, cells were grown in 10-well slides (Greiner) and imaged with a Visitron Live Spinning Disk microscope, using a Plan-Apochromat 63x/1.4 Oil DIC III objective, 488 nm diode (100 mW) and 561 nm DPSS (200 mW, AOTF-controlled) lasers. The system is attached to an EM-CCD back-illuminated evolve EM512 camera (512 × 512 pixel, 16 μm pixel size, frame transfer, 16-bit, appr. 34 FPS, QE 95%) and is equipped with environmental control for keeping constant temperature (37°C) and $CO_2$ concentration (5%). 0.1 μM DAPGreen (Dojindo) was added 90 min before starting imaging.

### HeLa and Hap1 cell culture and treatment

HeLa and Hap1 cells were grown at 37°C with constant 5% $CO_2$ supply and split at regular intervals. Both cell lines were regularly screened for mycoplasma infection. HeLa cells were grown in DMEM high glucose medium (Thermo Fisher Scientific), supplemented with 10% FBS and 1% penicillin/streptomycin. Hap1 cells were grown in IMDM medium (Thermo Fisher Scientific) supplemented with 10% FBS and 1% penicillin–streptomycin. Overexpression of RFP–LC3B and iRFP–Rab7 was performed for 24 h using Lipofect-amine 2000, according to the manufacturer's instructions. For the overexpression of RFP–LC3B in HeLa cells, DAPGreen (0.1 μM) was added 3 h before imaging, together with the proteasomal inhibitor MG-132 (10 μM). For staining lysosomes, LysoTracker Blue (Thermo Fisher Scientific) was added into Hap1 cells and DIV21 neurons at a final concentration of 50 nM. The cells were then washed once in PBS for removing the excess of dye just before live cell imaging.

---

**(F)** Quantification of the bands obtained on the Western blots (F). Puromycin band intensity values were normalised to the GAPDH loading control first and then adjusted to the T0 WT sample to get fold change values. The graphs represent the mean of the fold change ± SEM from n = 3 independent experiments.

## SDS–PAGE and Western blot analysis

Cells were washed three times with ice-cold PBS and gently scraped in lysis buffer: 50 mM Tris–HCl, pH 7.4, 1 mM EGTA, 1 mM EDTA, 1% Triton X-100, 0.27 M sucrose, 1 mM DTT, and complete protease inhibitors (Roche). Cell lysates were snap-frozen until use. Samples were spun at 10,000$g$ for 10 min at 4°C, and the supernatant containing the cell lysate was collected. Protein concentration was determined by Bradford assay using different concentrations of BSA as protein standards. SDS–PAGE and Western blot analyses were performed using standard protocols. Primary antibody incubation was always performed overnight at 4°C, whereas HRP-conjugated secondaries were incubated for 1 h at RT. Western blot images were acquired using a ChemiDoc gel imaging system (Bio-Rad). The following antibodies and dilutions were used in this study: rabbit anti-p62 (1:1,000; Abcam), mouse anti-NBR1 (1:1,000; Abnova 6B11), mouse anti-ubiquitin conjugates (1:1,000; Enzo FK2), mouse anti-LC3B (1:500; Nanotools), mouse anti-GAPDH (1:10,000; Sigma-Aldrich), mouse anti-tubulin III-beta (1:1,000; BioLegend TUJ1), mouse anti-puromycin (1:20,000; Merck 12D10), mouse anti-strep-tag (1:2,000; QIAGEN), mouse anti-RFP (1:1,000; ChromoTek), goat anti-rabbit HRP (1:10,000; Dianova), goat anti-mouse HRP (1:10,000; Dianova).

## Puromycin pulse-chase experiment

The experiments were performed as described by reference 13. One sample per cell line was pre-treated with 15 $\mu$M cycloheximide for 10 min as a negative control. Afterwards, 5 $\mu$g/ml puromycin was added to all of the samples, including the ones pre-treated with cycloheximide, and the treatment was performed for 2 h. At the end of this pulse time, samples were chased at different time points. The one pre-treated with cycloheximide was harvested immediately, along with one treated only with puromycin. The remaining ones were washed twice in PBS, re-incubated with standard growing medium, and lysed after 1, 3, 6, and 24 h. For the controls, mScarlet–p62 WT cells were treated for 6 h with 5 $\mu$g/ml puromycin alone or in combination with either 10 $\mu$M MG-132 or 400 nM bafilomycin or a combination of MG-132 and bafilomycin or 1 $\mu$M VPS34 IN1 (Biozol). One sample was pre-treated with 15 $\mu$M cycloheximide as a negative control for 10 min. At the end of the 6 h treatment time, cells were harvested and lysed.

## Microscopy image analysis

Images were processed and analysed using ImageJ/Fiji. For counting p62 and DAPGreen particle number and size a threshold was set and validated manually for each channel. In the case of image analyses of neurons, a different threshold was set for the cell body and for the processes. Every replicate within the same experiment was analysed using the same threshold. The function "Analyze particles" applied to the threshold-derived puncta and excluding all of the particles smaller than 0.5 $\mu$m$^2$ was used for the quantifications. For the colocalisation analysis, the particles identified in the first channel via the threshold function, as described above, were added to the region of interest and copied into the other channel. All of the region of interests that contained some signal over the threshold in the second channel were counted as colocalising particles.

## Western blot data analysis

Western blot data were analysed with Fiji, using the gel-plotting function, according to the manual. A rectangular square was drawn around the bands and/or lanes of interest. The function "plot lanes" was used for generating the graph corresponding to the band intensity. The peaks in the graph represent each band and the area of the peaks was measured for obtaining the intensity.

## Microscopy-based pull-down assay

GST and GST-4x ubiquitin chains were purified as previously described (24). Purified proteins were diluted in washing buffer: 25 mM HEPES, 150 mM NaCl, and 1 mM DTT. Glutathione Sepharose 4B beads (GE Healthcare) were incubated with either GST or GST-4x ubiquitin (4 $\mu$g of purified protein per each $\mu$l of dry beads) in washing buffer. The mixture was incubated for 1 h at 4°C. Afterwards, the beads were washed once in washing buffer and added on top of cell lysates in a 96-well glass-bottom microscopy plate (Greiner). About 100 $\mu$g of lysate were incubated with 7 $\mu$l of beads (1:1 slurry) in a final volume of 25 $\mu$l. After 30 min of incubation at RT, the samples were imaged with a Zeiss LSM 980 microscope using a 40x Plan-Apochromat 1.2 Imm. Corr. DIC (water immersion) WD 0.41 mm objective and a 561 nm (25 mW) laser.

## Statistical analysis

All the statistical tests and graphs were generated with Prism 9. $T$ tests and two-way ANOVAs were calculated using the standard programme settings applied to "column" or "grouped data." Details about statistical tests are shown in the figure legends.

# Supplementary Information

# Acknowledgements

We thank the Max Perutz Labs BioOptics Facility for technical support. This work was supported by the ERC Grant No. 646653 and the Austrian Science Fund Grant No. F79 (S Martens). We thank Dr Luca Ferrari for critical reading of the manuscript.

## Author Contributions

R Trapannone: conceptualization, data curation, formal analysis, validation, investigation, visualization, methodology, and writing—original draft, review, and editing.
J Romanov: investigation, methodology, and writing—review and editing.
S Martens: conceptualization, resources, supervision, funding acquisition, project administration, and writing—original draft, review, and editing.

## Conflict of Interest Statement

S Martens is member of the scientific advisory board of Casma Therapeutics.

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
