## [Reviewer comments · Life Science Alliance]

Life Science Alliance

p62 and NBR1 functions are dispensable for aggregophagy in mouse ESCs and ESC-derived neurons

Riccardo Trapannone, Julia Romanov, and Sascha Martens

DOI: <https://doi.org/10.26508/lsa.202301936>

Corresponding author(s): Sascha Martens, University of Vienna and Riccardo Trapannone, University of Vienna

Review Timeline:

Submission Date:	2023-01-20
Editorial Decision:	2023-02-22
Revision Received:	2023-07-19
Editorial Decision:	2023-08-08
Revision Received:	2023-08-10
Accepted:	2023-08-11

Scientific Editor: Novella Guidi

Transaction Report:

February 22, 2023

Re: Life Science Alliance manuscript #LSA-2023-01936

Prof. Sascha Martens
University of Vienna
Max Perutz Labs / Vienna Biocenter / University of Vienna
Dr. Bohrgasse 9/5
Vienna 1030
Austria

Dear Dr. Martens,

Thank you for submitting your manuscript entitled "p62 and NBR1 functions are dispensable for aggregophagy in mouse ESCs and ESC-derived neurons" to Life Science Alliance. The manuscript was assessed by expert reviewers, whose comments are appended to this letter. We invite you to submit a revised manuscript addressing the Reviewer comments.

Thank you for this interesting contribution to Life Science Alliance. We are looking forward to receiving your revised manuscript.

Sincerely,

B. MANUSCRIPT ORGANIZATION AND FORMATTING:

Reviewer #1 (Comments to the Authors (Required)):

In the present manuscript, the authors described the functional redundancy of two autophagy receptors, p62 and NBR1, in murine ESCs as well as ESCs derived neurons. The main message is that protein aggregates clearance, via autophagy, is not driven by a single receptor but it is rather a cooperation and compensative feedback that involves several autophagy receptors. Therefore, targeting a single molecule could not be sufficient to have a biological effect.

The experiments are properly done, results are clear and also the employed technology is innovative. However, still remain an open question of how aggrephagy is regulated in this cells. Authors excluded the role of p62 and NBR1, individually and in combination, but they didn't elucidate the possible involvement of other receptors like Optineurin, TAX1BP1, NDP52. In other type of selective autophagy, like mitophagy or ERphagy, several receptors are involved and they somehow cooperate/compensate. Maybe the authors should consider to address this point knocking out/down, together with p62, other receptors. This may help to understand if p62 is important in association with someone else than NBR1 or if knockout of multiple receptors is demanded.

Minor point

- Fig. S1 and Fig. S2 can be merged
- Fig. 2A. authors should better explain and maybe quantify the "diffuse cytoplasmic localisation" of p62 puncta
- Fig. S5 maybe provide a graph and statistic
- Fig. 7C needs statistic and should be before Fig. 7B accordingly to the narrative
- Fig. S9 graph needs statistic

Some parts of the text are a bit complicated to read, maybe authors could adjust and make the grammar easier.

Reviewer #2 (Comments to the Authors (Required)):

Selective autophagy is the major way to clear toxic protein aggregates in cells. Two main aggrephagy receptors p62 and NBR1 are important for degradation of aggregates in immortalised cell lines. In the current work, the authors used mouse ESCs to explore the function of the p62 and NBR1 and came to a different conclusion. The work inspired that other receptors or unknown factors may take the place of p62 and NBR1 to clear protein aggregates in mouse ESCs and ESC-derived neurons. In general, it is a nice work and the conclusions are convincing. I have a few questions and several points need to be improved.

1. In Figure S2B and Figure 3B, the turnover of mScarlet-p62 WT seemed deficiency, even compared to mScarlet-p62 LIR mutant, let alone the non-tagged p62. I wondered whether the mScarlet-p62 still a functional autophagy receptor in the cell line. The author should confirm this point. For example, the interaction between mScarlet-p62 WT/LIR mutant with LC3 should be confirmed.
2. It was reported that the UBA domain is required for polyubiquitin chain-induced phase separation of p62(PMID: 29507397). Did the mScarlet-p62 F406V condensate have liquidity?
3. Does p62 and NBR1 deficiency in ESCs affect the degradation of toxic protein aggregates caused by aggregation-prone proteins such as such as Huntingtin and Tau?
4. Why the lipidated LC3B was obviously stronger in first lane?
5. The form of P-value on bar graphs should be consistent.
6. The references should be checked carefully. "In addition, a CCT2 dependent macroautophagy pathway may compensate for the loss of p62 and NBR1 function [27]" in Page 10 was cited inaccurately.

Reviewer #3 (Comments to the Authors (Required)):

Trapanone, Romanov and Martens conclude with the title of their present manuscript that "p62 and NBR1 functions are dispensable for aggrephagy in mouse ESCs and ESC-derived neurons», which at first glance may seem a surprising finding. However, on second thought it should not be surprising that aggrephagy is a robust process with a number of compensating

mechanisms.

Generally, I find it very interesting to see results on aggrephagy studies in ESC and differentiated neurons using endogenous tagged p62/SQSTM1 WT, LIR mutant and UBA mutant. The experimental strategy with KI of mScarlet is very good and the experiments are well performed and the data quantified to allow comparisons and valid conclusions. The relevant literature is very well referenced and the Discussion is balanced.

1. It is an important observation that LIR mutant p62 is degraded by autophagy. Degradation of LIR mutant p62 shows that at normal levels of p62 there are compensating backup mechanisms at work. NBR1 is one of the compensating factors. The authors should perhaps show immunostaining of NBR1 to see if there is actual recruitment of NBR1 to p62 bodies.

2. The authors employ the compound DAPGreen for live cell imaging in this study as mentioned in their own words : "In order to visualise the degree of colocalization of p62 with forming autophagosomes, we stained the cells with DAPGreen, a compound that emits fluorescence when it gets incorporated into autophagosomes and lysosomes [31]." Although it is referred to a paper here (ref 31) I think the use of this compound is so central to this study and so far very little used in the field. It is therefore important to show colocalization studies of DAPGreen here in these cell systems (ESC and ESC-derived neurons) with markers like WIPI2 and LC3B for autophagosomes and LAMP1 or LAMP2 for late endosomes and lysosomes.

3. In this last sentence in the INTRODUCTION a word is missing "From this comparison we showed that loss of function of the two cargo receptors p62 and NBR1 does not the clearance of protein aggregates." ... The missing word may be "block".

4. Middle of the first page in the Results section. "Live cell imaging was performed so that p62- and DAPGreen-positive structures could be simultaneously observed (Fig. S3)". The figure reference is incorrect.

5. In the second last page of the DISCUSSION this sentence ends with a wrong reference. "In addition, a CCT2 dependent macroautophagy pathway may compensate for the loss of p62 and NBR1 function [27]." The correct reference should be this paper on CCT2-mediated aggrephagy: PMID: 35366418.

6. It is relevant to also blot for p62 as a control in the puromycin experiments shown in Fig. 8.

We thank the reviewers for their constructive comments, which we have taken very seriously. Please find below a point-by-point reply to each of the comments raised.

Reviewer #1 (Comments to the Authors (Required)):

In the present manuscript, the authors described the functional redundancy of two autophagy receptors, p62 and NBR1, in murine ESCs as well as ESCs derived neurons. The main message is that protein aggregates clearance, via autophagy, is not driven by a single receptor but it is rather a cooperation and compensative feedback that involves several autophagy receptors. Therefore, targeting a single molecule could not be sufficient to have a biological effect. The experiments are properly done, results are clear and also the employed technology is innovative. However, still remain an open question of how aggrephagy is regulated in this cells. Authors excluded the role of p62 and NBR1, individually and in combination, but they didn't elucidate the possible involvement of other receptors like Optineurin, TAX1BP1, NDP52. In other type of selective autophagy, like mitophagy or ERphagy, several receptors are involved and they somehow cooperate/compensate. Maybe the authors should consider to address this point knocking out/down, together with p62, other receptors. This may help to understand if p62 is important in association with someone else than NBR1 or if knockout of multiple receptors is demanded.

This is an excellent point. Indeed, we believe that some other cargo receptors might be involved in compensating the loss of function of p62 and NBR1. In response to the reviewer's comment, we investigated whether TAX1BP1 is involved. TAX1BP1 has been shown to play an important role in aggrephagy. Therefore, we transfected a pool of 4 siRNAs to knock down TAX1BP1 in WT and p62/NBR1-deficient cells. The transfection was performed for 48 hours and afterwards cells were treated as in Figure S9 (Cycloheximide in combination with Puromycin, Puromycin alone and Puromycin combined with VPS34 IN1). The treatment was performed for 6 hours and afterwards cell lysates were analyzed by Western blotting. Although TAX1BP1 levels were reduced quite efficiently by the knock down, p62 levels also seemed to be lowered in the TAX1BP1 siRNA samples. The amount of puromycin-conjugated peptides appears to be lower upon TAX1BP1 knock down and importantly still decreased even in the p62 UBA mutant/NBR1 knock cell line, suggesting that TAX1BP1 is not essential. Taken together, these preliminary data are still difficult to interpret without extensive controls. We therefore refrained from including these data in the manuscript but show them below (Figure 1). In general, we believe that knocking out other cargo receptors via CRISPR/Cas9 in various combinations would be a better way to tackle the question of the interplay of these additional receptors in aggrephagy, but this is outside the scope of this study.

Figure 1: TAX1BP1 knock down in mScarlet-p62 WT and mScarlet-p62 F408V/NBR1 KO ESCs. Cells were transfected with either negative control siRNA or a pool of 4 TAX1BP1-specific siRNAs for 48h. Afterwards the indicated treatment were performed.

Minor point

- Fig. S1 and Fig. S2 can be merged

Thank you. Figures S1 and S2 have been merged. Because Figure S3 is also conceptually related, we merged this into the new S1 figure as well.

- Fig. 2A. authors should better explain and maybe quantify the "diffuse cytoplasmic localisation" of p62 puncta

The text has been corrected with a clearer explanation (lines 139-141). We also added an additional panel to Fig. S3 (B) showing two neuronal cell bodies and two pluripotent stem cells either WT or LIR mutant. We used the "Fire" lookup table so that the cytoplasmic background signal is more visible.

- Fig. S5 maybe provide a graph and statistic

Statistics could not be performed in neurons in the time available for the revision, but we repeated the experiments in ES cells and performed the statistics in this system. A graph has been added to the figure (now Fig. S4)

- Fig. 7C needs statistic and should be before Fig. 7B accordingly to the narrative

The panels have been now been rearranged according to the suggestion. Statistics have been performed as explained in the figure legend. Non-significant changes with P values above 0.05 are not shown in the graphs.

- Fig. S9 graph needs statistic

The experiment, originally performed as a single replicate, has now been repeated and statistics have been performed. The figure (now Fig. S9) has been updated.

Some parts of the text are a bit complicated to read, maybe authors could adjust and make the grammar easier.

Thank you. Long sentences were shortened and we made some parts clearer and easier to read.

Reviewer #2 (Comments to the Authors (Required)):

Selective autophagy is the major way to clear toxic protein aggregates in cells. Two main autophagy receptors p62 and NBR1 are important for degradation of aggregates in immortalised cell lines. In the current work, the authors used mouse ESCs to explore the function of the p62 and NBR1 and came to a different conclusion. The work inspired that other receptors or unknown factors may take the place of p62 and NBR1 to clear protein aggregates in mouse ESCs and ESC-derived neurons. In general, it is a nice work and the conclusions are convincing. I have a few questions and several points need to be improved.

1. In Figure S2B and Figure 3B, the turnover of mScarlet-p62 WT seemed deficiency, even compared to mScarlet-p62 LIR mutant, let alone the non-tagged p62. I wondered whether the mScarlet-p62 still a functional autophagy receptor in the cell line. The author should confirm this point. For example, the interaction between mScarlet-p62 WT/LIR mutant with LC3 should be confirmed.

Thank you. This is a very good point. In one of our previous publications (Wurzer et al. 2015) we have shown that both mCherry-p62 WT and LIR mutant are able to bind LC3 *in vitro*. We now added this reference into the manuscript.

2. It was reported that the UBA domain is required for polyubiquitin chain-induced phase separation of p62 (PMID: 29507397). Did the mScarlet-p62 F406V condensate have liquidity?

This is a very interesting point. FRAP experiments would be the ideal approach to tackle this question. The ESCs are too small and the p62 positive structures are too dense for meaningful FRAP experiments. We have attempted FRAP in neurons. However, due to fact that we work with genomically tagged proteins rather than overexpression, the condensates are very small. Moreover, the condensates are very mobile and therefore it is technically almost impossible to track them after bleaching. The high laser intensity for the bleaching also resulted in neuronal cell death. For this reason, we could not add these experiments to the manuscript.

3. Does p62 and NBR1 deficiency in ESCs affect the degradation of toxic protein aggregates caused by aggregation-prone proteins such as such as Huntingtin and Tau?

We have made serious efforts to implement the AgDD system (<https://pubmed.ncbi.nlm.nih.gov/27229621/>) in the mESCs and the derived neurons. While this system worked well in HeLa cells, the expression of AgDD was consistently very low in the mESCs and the derived neurons such that no aggregates were formed. For this reason, we employed the puromycin-induced protein aggregation protocol to follow the removal of puromycin-conjugated peptides, which has the advantage of monitoring endogenous proteins.

4. Why the lipidated LC3B was obviously stronger in first lane?

Thanks for pointing this out. We assume you are referring to the blot in Figure 7. We observed several times that basal autophagy (and therefore the basal levels of lipidated LC3B) are stronger in WT cells compared to any of the mutant tested. This indicates that basal autophagy is slightly higher in the WT cells compared to the p62 and NBR1 mutants and would be consistent with these two proteins being drivers of basal autophagy. However, this difference did not translate into major accumulations of the two cargo receptors with the exception of the p62 UBA mutant/NBR1 knockout cell line (Figure 7 A,B).

5. The form of P-value on bar graphs should be consistent.

In order to make the interpretation easier, we only show P-values when the differences are significant. All the "n.s" labels have been removed as explained in the figure legends except in those cases when the p-value is slightly above the threshold of 0.05.

6. The references should be checked carefully. "In addition, a CCT2 dependent macroautophagy pathway may compensate for the loss of p62 and NBR1 function [27]" in Page 10 was cited inaccurately.

Thank you. The right reference has now been cited.

Reviewer #3 (Comments to the Authors (Required)):

Trapannone, Romanov and Martens conclude with the title of their present manuscript that "p62 and NBR1 functions are dispensable for aggrephagy in mouse ESCs and ESC-derived neurons», which at first glance may seem a surprising finding. However, on second thought it should not be surprising that aggrephagy is a robust process with a number of compensating mechanisms.

Generally, I find it very interesting to see results on aggrephagy studies in ESC and differentiated neurons using endogenous tagged p62/SQSTM1 WT, LIR mutant and UBA mutant. The experimental strategy with KI of mScarlet is very good and the experiments are well performed and the data quantified to allow comparisons and valid conclusions. The relevant literature is very well referenced and the Discussion is balanced.

1. It is an important observation that LIR mutant p62 is degraded by autophagy. Degradation of LIR mutant p62 shows that at normal levels of p62 there are compensating backup mechanisms at work. NBR1 is one of the compensating factors. The authors should perhaps show immunostaining of NBR1 to see if there is actual recruitment of NBR1 to p62 bodies.

This is a very important point. Although immunostaining of p62 condensates is not as informative as live cell imaging, due to the loss of some of these structures upon fixation, we tested the colocalisation of p62 and NBR1. We see some puncta where p62 and NBR1 colocalize both in the WT and in the LIR mutant ESC lines. Nevertheless, we also see some NBR1 puncta in the NBR1 KO negative control. Therefore, we cannot be absolutely sure that the signal observed is specific and if the reviewer agrees, we prefer not to publish these data in the manuscript. The results are displayed below (Figure 2).

Figure 2: Immunostaining of the indicated ESC lines. Cells were stained with anti-NBR1 and anti-p62 antibodies. Scale bar: 20 μm . 5x magnification of the selected areas is shown.

2. The authors employ the compound DAPGreen for live cell imaging in this study as mentioned in their own words : "In order to visualise the degree of colocalization of p62 with forming autophagosomes, we stained the cells with DAPGreen, a compound that emits fluorescence when it gets incorporated into autophagosomes and lysosomes [31]." Although it is referred to a paper here (ref 31) I think the use of this compound is so central to this study and so far very little used in the field. It is therefore important to show colocalization studies of DAPGreen here in these cell systems (ESC and ESC-derived neurons) with markers like WIPI2 and LC3B for autophagosomes and LAMP1 or LAMP2 for late endosomes and lysosomes.

We agree that it's important to show our own data about the DAPGreen compound. We performed some colocalization experiments in HeLa cells and Hap1 cells. The results are shown in Figure S2. We overexpressed RFP-tagged LC3 in HeLa cells and showed the degree of colocalization with DAPGreen. We also expressed iRFP-Rab7 in Hap1 cells to check the colocalization with endosomes. Finally, we used LysoTracker in Hap1 cells and in mScarlet-p62 WT neurons to check the colocalization with lysosomes.

3. In this last sentence in the INTRODUCTION a word is missing "From this comparison we showed that loss of function of the two cargo receptors p62 and NBR1 does not the clearance of protein aggregates." ... The missing word may be "block".

Yes exactly. The sentence has been corrected.

4. Middle of the first page in the Results section. "Live cell imaging was performed so that p62- and DAPGreen-positive structures could be simultaneously observed (Fig. S3)". The figure reference is incorrect.

Thank you. The reference to the figure is now correct.

5. In the second last page of the DISCUSSION this sentence ends with a wrong reference. "In addition, a CCT2 dependent macroautophagy pathway may compensate for the loss of p62 and NBR1 function [27]." The correct reference should be this paper on CCT2-mediated aggrephagy: PMID: 35366418.

Thank you. The right reference has now been cited.

6. It is relevant to also blot for p62 as a control in the puromycin experiments shown in Fig. 8.

p62 blot has been performed on the Puromycin pulse-chase experiment. The samples were not the same as displayed in Fig. 8, although they correspond to one the replicates used for the quantification. For this reason, the blots are shown in a separate figure (Fig. S8).

August 8, 2023

RE: Life Science Alliance Manuscript #LSA-2023-01936R

Prof. Sascha Martens
University of Vienna
Max Perutz Labs / Vienna Biocenter / University of Vienna
Dr. Bohrgasse 9/5
Vienna 1030
Austria

Dear Dr. Martens,

Thank you for submitting your revised manuscript entitled "p62 and NBR1 functions are dispensable for aggregophagy in mouse ESCs and ESC-derived neurons". We would be happy to publish your paper in Life Science Alliance pending final revisions necessary to meet our formatting guidelines.

-please add the Twitter handle of your host institute/organization as well as your own or/and one of the authors in our system
-please add a callout for Fig 2B, Fig S1G, Fig S2C, Fig S2D, Fig S4A B C, Fig S8 A B C D E, Fig S9 A B to your main manuscript text

A. FINAL FILES:

B. MANUSCRIPT ORGANIZATION AND FORMATTING:

**Submission of a paper that does not conform to Life Science Alliance guidelines will delay the acceptance of your

manuscript.**

The license to publish form must be signed before your manuscript can be sent to production. A link to the electronic license to publish form will be sent to the corresponding author only. Please take a moment to check your funder requirements.

Sincerely,

Reviewer #1 (Comments to the Authors (Required)):

Authors addressed the referee's questions

Reviewer #2 (Comments to the Authors (Required)):

The authors have addressed my concerns.

Reviewer #3 (Comments to the Authors (Required)):

The authors have answered my comments in a satisfactory manner. I recommend publication of the revised manuscript.

August 11, 2023

RE: Life Science Alliance Manuscript #LSA-2023-01936RR

Prof. Sascha Martens
University of Vienna
Max Perutz Labs / Vienna Biocenter / University of Vienna
Dr. Bohrgasse 9/5
Vienna 1030
Austria

Dear Dr. Martens,

Thank you for submitting your Research Article entitled "p62 and NBR1 functions are dispensable for aggregophagy in mouse ESCs and ESC-derived neurons". It is a pleasure to let you know that your manuscript is now accepted for publication in Life Science Alliance. Congratulations on this interesting work.

DISTRIBUTION OF MATERIALS:

Again, congratulations on a very nice paper. I hope you found the review process to be constructive and are pleased with how the manuscript was handled editorially. We look forward to future exciting submissions from your lab.

Sincerely,
